# Drug resistance mechanisms create targetable proteostatic vulnerabilities in Her2 + breast cancers

Navneet Singh[1¤a], Lindsey Romick-Rosendale[2], Miki Watanabe-Chailland©[2], Lisa M. Privette Vinnedge©[2,3]*, Kakajan Komurov©[1,4¤b]*

**1** Division of Experimental Hematology and Cancer Biology, Cancer and Blood Diseases Institute, Cincinnati Children's Hospital Medical Center, Cincinnati, OH, United States of America, **2** Division of Pathology and Laboratory Medicine, Cincinnati Children's Hospital Medical Center, Cincinnati, OH, United States of America, **3** Division of Oncology, Cancer and Blood Diseases Institute, Cincinnati Children's Hospital Medical Center, Cincinnati, OH, United States of America, **4** Department of Pediatrics, University of Cincinnati College of Medicine, Cincinnati, OH, United States of America

¤a Current address: Case Comprehensive Cancer Center, Case Western Reserve University, Cleveland, OH, United States of America
¤b Current address: Champions Oncology Inc, Hackensack NJ, United States of America
* komurov@hotmail.com (KK); lisa.privette@cchmc.org (LMPV)

**Data Availability Statement:** RNAseq data are available in the S2 Table.

**Funding:** This work was supported by NIH awards R01CA193549 (KK and LMPV), R37CA218072

## Abstract

Oncogenic kinase inhibitors show short-lived responses in the clinic due to high rate of acquired resistance. We previously showed that pharmacologically exploiting oncogene-induced proteotoxic stress can be a viable alternative to oncogene-targeted therapy. Here, we performed extensive analyses of the transcriptomic, metabolomic and proteostatic perturbations during the course of treatment of Her2+ breast cancer cells with a Her2 inhibitor covering the drug response, resistance, relapse and drug withdrawal phases. We found that acute Her2 inhibition, in addition to blocking mitogenic signaling, leads to significant decline in the glucose uptake, and shutdown of glycolysis and of global protein synthesis. During prolonged therapy, compensatory overexpression of Her3 allows for the reactivation of mitogenic signaling pathways, but fails to re-engage the glucose uptake and glycolysis, resulting in proteotoxic ER stress, which maintains the protein synthesis block and growth inhibition. Her3-mediated cell proliferation under ER stress during prolonged Her2 inhibition is enabled due to the overexpression of the eIF2 phosphatase GADD34, which uncouples protein synthesis block from the ER stress response to allow for active cell growth. We show that this imbalance in the mitogenic and proteostatic signaling created during the acquired resistance to anti-Her2 therapy imposes a specific vulnerability to the inhibition of the endoplasmic reticulum quality control machinery. The latter is more pronounced in the drug withdrawal phase, where the de-inhibition of Her2 creates an acute surge in the downstream signaling pathways and exacerbates the proteostatic imbalance. Therefore, the acquired resistance mechanisms to oncogenic kinase inhibitors may create secondary vulnerabilities that could be exploited in the clinic.

(LMPV), and a Department of Defense Breast Cancer Research Program level I award W81XWH-16-1-0028 (NS). We would like to acknowledge the assistance of the Research Flow Cytometry Core in the Division of Rheumatology and the NMR-based Metabolomics Core Facility at Cincinnati Children's Hospital Medical Center. The funders had no role in study design, data collection and analysis, decision to publish, or preparation of the manuscript.

**Competing interests:** KK is an employee of Champions Oncology Inc, and holds stocks there and at Pfizer Inc. This does not alter our adherence to PLOS ONE policies on sharing data and materials.

## Introduction

The Her2 (*ERBB2*) receptor tyrosine kinase is amplified in 15–20% of breast cancers, and historically has correlated with poor prognosis. Her2 is a member of the EGFR (epidermal growth factor receptor) family of receptor tyrosine kinases, which also includes EGFR (ErbB1), Her3 (ErbB3), and Her4 (ErbB4) [1]. The EGFR family of receptors is activated by ligand binding and subsequent dimerization, which leads to the activation of downstream pathways most often associated with mitogenic and pro-survival signaling. Some of the best-characterized of these signaling pathways include the PI3K/AKT/mTOR cascade, which promotes pro-survival signaling and protein synthesis, and the Ras/MAPK pathway that promotes cellular migration and cell cycle progression [2–4]. As such, this family of receptors is often the target of genetic alterations in cancers that result in their constitutive activation: e.g. *EGFR* is frequently mutated in lung cancers and amplified in gliomas, while *ERBB2* (Her2) is frequently amplified in breast cancers.

Clinical management of Her2+ breast cancers includes Her2-targeted monoclonal antibody (mAb) trastuzumab combined with chemotherapy, followed by, or lately in combination with, the newer generation of Her2-targeted mAb pertuzumab. These are followed by trastuzumab-emtansine, an antibody-drug conjugate, at later treatment stages, or small molecule inhibitors of EGFR/Her2, such as lapatinib. These Her2-targeted therapies have dramatically altered the outcomes for Her2+ breast cancer patients, especially in early disease. However, the metastatic Her2+ breast cancer is still an incurable disease, and all of these patients inevitably relapse on anti-Her2 therapies [5]. Therefore, identifying the mechanisms of acquired resistance to anti-Her2 therapy, and developing novel therapeutic targeting strategies within the relapsed setting is a high priority goal.

The bulk of research effort in the acquired drug resistance field has focused on the alternative oncogenic bypass mechanisms and their potential targeting to prevent resistance. However, oncogenic hyperactivation, in addition to forcing cell growth and division, also triggers multiple homeostatic stress checkpoints such as DNA damage response, metabolic stress and proteotoxic stress, which could present opportunities for therapeutic exploitation [6, 7]. Although the traditional approach to cancer targeted therapy focused on inhibiting the driver oncogene, pharmacological forcing of irremediable oncogenic stress has been suggested as a viable alternative, especially in the cancers where oncogene-targeted therapy is not feasible (e.g. *MYC*-driven cancers) or where tumors have gained resistance to the oncogene-targeting agent [8–11]. We have previously shown that strong oncogenic signaling through Her2 amplification imposes a proteotoxic stress on the mammary epithelial cell that has to be mitigated by the activation of compensatory stress relief systems to allow for the tumor cell to survive [12]. Her2 + breast cancer cells are characterized by increased protein synthesis load due to chromosomal amplifications and hyperactive Her2/mTOR signaling, which creates dependence on the endoplasmic reticulum (ER)-associated degradation (ERAD) pathway to maintain protein homeostasis and prevent proteotoxic stress [12]. Thus, pharmacologic inhibition of ERAD through targeting of its central player, the p97 VCP ATPase, led to oncogenic Her2-dependent proteotoxic stress and cell death [12]. Although our study provided strong rationale for targeting of ERAD in Her2+ breast cancers, it is unknown how protein homeostasis and the associated dependencies change after prolonged anti-Her2 therapies. Since new treatment modalities, such as ERAD targeting, for Her2+ breast cancer patients are likely to enter the clinic in the heavily pre-treated patient populations, it is important that we understand the signaling and proteostasis dynamics during the process of cellular adaptation to anti-Her2 therapy.

To address this goal, in this study, we developed a model of acquired resistance to anti-Her2 therapy in Her2+ breast cancer cells by employing a frequently used strategy of *in vitro* dose escalation. In line with previous reports, we found that the resistance to Her2 inhibition

is associated with the compensatory overexpression of Her3 and the Her2-independent re-activation of downstream mitogenic pathways. However, acute Her2 inhibition leads to the metabolic and proteotoxic stress, and subsequent protein synthesis block due to PERK-mediated phosphorylation of the translation initiation factor eIF2α. Interestingly, the compensatory overexpression of Her3 is unable to mitigate the ER stress due to Her2 inhibition, and therefore necessitates the overexpression of the protein phosphatase 1 subunit GADD34 (*PPP1R15A*) to relieve the ER stress-induced block to protein synthesis and promote Her2-independent cell proliferation. Strikingly, while GADD34-mediated uncoupling of ER stress from protein synthesis block allows for active cell growth under Her2 inhibition, it also imposes greater dependence on the ER quality control machinery to clear the proteotoxic aggregates and promote cell survival. Accordingly, Her2+ cells at the acquired resistance stage to Her2 inhibition are hypersensitive to the pharmacologic and genetic targeting of ERAD due to unresolved ER stress and proteotoxic load. Our studies provide strong rationale for the consideration of ERAD-targeted therapies in Her2+ breast cancer patients who have progressed on prior Her2-targeted therapies. More generally, this study also supports the notion of identifying and targeting the secondary vulnerabilities (i.e. collateral sensitivities) imposed by the drug resistant state in cancers [13–16].

## Materials and methods

### Cell culture

Cell lines were purchased from the American Type Culture Collection (ATCC). Human HER2 + breast cancer SKBR3 (HTB-30) cells were cultured in RPMI 1640 (Gibco) containing 10% fetal bovine serum with 0.1% antibiotic and antimycotic (Gibco). Human mammary epithelial MCF10A (CRL-10317) cells were cultured in Dulbecco's modified Eagle's medium/F12 containing 10% horse serum with 0.1% antibiotic and antimycotic (Gibco), hydrocortisone, cholera toxin, insulin (all from Sigma) and EGF (PeproTech Inc.). For drug treatments, cells were incubated with lapatinib was from (Selleck Chemicals, S1028), 250nM CB-5083 (Cayman Chemicals, 19311), and 15µM guanabenz (Tocris, 0885).

### Lapatinib drug treatment

Lapatinib-resistant cell lines were generated by chronic exposure of 250nM lapatinib. Media was refreshed every two days with fresh lapatinib. Every two months, lapatinib concentration was increased from 250nm to 500nM, followed by 500nM to 1uM, then the cells were maintained in 1uM lapatinib. Viability analyses (growth rates) and western blotting assays were performed every two months before increasing the concentration of lapatinib. Lapatinib resistance was confirmed by inhibition of phosphorylation of HER2 expression in western blot.

### Lentiviral constructs and transfections

HER2 and HER3 expression plasmids were purchased from Addgene. The shVCP (TRC0000004249) and shHER3 (TRCN0000218392) pLKO.1 constructs were from the Mission shRNA collection from Sigma-Aldrich. Lentivirus particles expressing shRNA against the gene of interest were generated by co-transfection with the VSV-G packaging and CDNL envelope plasmids (courtesy of Biplab Dasgupta, Cincinnati Children's Hospital Medical Center, Cincinnati, OH) into HEK-293T cells using jetPRIME transfection reagent. Lentiviral supernatant was collected every 24 hours after transfection for 3 days. Cells were infected with lentiviral supernatant in the presence of polybrene (Sigma). shHER3 cells were selected in puromycin before analysis.

## Western blotting

Total cellular proteins were extracted using RIPA buffer, separated on an SDS-PAGE gel, and electrophoretically transferred onto PVDF membrane. The membranes were blocked in 5% dry milk in tris-buffered saline-Tween 20 for 1 hour. Blocked membrane were probed with primary antibodies (1:1000) overnight (S1 Table) in 5% bovine serum albumin. B-actin was used as a loading control. Membranes were incubated with secondary antibody (1:5000) and visualized using a gel imager (Azure Biosystems).

## RT-PCR

Total RNA extracted from the cells using Tri reagent (Sigma) as described previously [PMID: 23055106]. RevertAid First-strand cDNA synthesis kit (Fermentas Life Sciences, Glen Burnie, MD) was used to reverse transcribe total RNA. Gene expression levels were measured by real-time RT-PCR using the SYBER green PCR amplification kit according to the manufacturer's protocol (Promega, Madison, WI). With specific primers for the target genes. B actin was used as a housekeeping gene for equal loading of RNA, and relative fold expression was calculated.

Error bars represent the standard deviation. Primer sequences used to amplify target genes were as follows:

Actin: (Forward Primer) tcctctcccaagtccacacagg and (Reverse Primer) gggcacgaaggctcatcattc

GADD34: (Forward Primer) tccgactgcaaaggcggctca (Reverse Primer): cagccaggaaatggacagtgac

## Cell viability analyses

Equal number of cells were seeded into 96 –well culture plates and incubated overnight then treated with drugs or lentivirus as indicated. After 72 hours, dead cells were removed from the plates by washing with PBS buffer and the attached cells were stained and fixed with crystal violet (Sigma) for 30 min at room temperature. After 30 min, excess stains was removed with tap water and the plates were dried at room temperature. Once dried, crystal violet crystals were re-dissolved in triton (Amresco), and cell density was determined by measuring the absorbance at 570nM in a microplate reader (Bio-tek Instruments).

## Glucose uptake kit

The glucose uptake colorimetric assay kit (K676–100, Biovision, CA) was used according to the manufacturer's instructions. Briefly, $10^4$ cells were seeded into a 96-well plate overnight. Cells were washed twice with PBS and starved in 100ul of serum free medium for 2 hours (to increase glucose uptake), then rewashed three times with PBS. The cells were starved or not starved for glucose by preincubating with 100 μL Krebs Ringer Phosphate HEPES (KRPH) buffer containing 2% BSA for 40 min. Cells were stimulated with or without insulin (1 μM) for 20 min to activate glucose transporter, and 10 μL of 10 mM 2-deoxyglucose (2-DG) was added and incubated for 20 min. The glucose uptake was measured by the cellular fluorescence (Ex/Em = 535/587 nm) in a microplate reader (BioTake, USA)

## Click it (protein synthesis assay)

Newly synthesized proteins were detected and measured with the click-it HPG kit (C10428 Thermo fisher Scientific; C10428) according to the manufacturer's instructions. Briefly, equal number of cells (50,000) were seeded on coverslips (for microscopy) and in 6 well plates (for flow cytometry) overnight. Media was removed and 250 μl of 50uM of click–it HPG solution

were added per well. Cells were incubated in 5% $CO_2$ humidified incubator for 30 minutes then cells were washed with PBS. For flow cytometry analysis, cells were trypsinized pelleted. For microscopy, cells remained on coverslips and were fixed with 4% formaldehyde at room temperature for 30 minutes then permeabilized with methanol for 1 hour at room temperature. All samples were then washed twice with PBS then 500 μL of click-iT reaction cocktail was added and incubated for 30 minutes in dark. Excess staining solution was removed and cells were washed with click-iT reaction rinse buffer. Finally, coverslips were mounted and analyzed using a Nikon A1R microscope wavelength or resuspended in PBS for flow cytometry analysis with a FACS Aria (BD Biosciences).

## Metabolomics

**Cell extraction.** The cell pellets were resuspended in 1.5mL ice-cold 80% methanol, vortex for 1 min and incubated on ice for 10min. The samples were centrifuged at 10,000x $g_n$ for 10 min at 4˚C. The supernatant, i.e. the polar extract, was dried in a SpeedVac centrifuge for 4–6 h and stored at -20˚C until further preparation for NMR data collection. On the day of the data collection, the dried hydrophilic cell extract samples are resuspended in 220 μL of NMR buffer (100 mM potassium phosphate (pH 7.3), 0.1% sodium azide, 1mM trimethylsilylproprionate (TSP) in 100% $D_2O$). The protein pellets were rinsed with 0.5 mL 80% methanol, centrifuged for 20min at 10,000x $g_n$ and dried in SpeedVac centrifuge for 1 h before stored in -80C freezer.

**Media sample processing.** On the day of the data collection, samples were thawed on ice and centrifuged 4000x $g_n$ for 5 min at 4˚C. The 550 $\mu L$ supernatant of media samples were aliquoted onto pre-washed 3 kDa spin filters (NANOSEP 3K, Pall Life Sciences), and centrifuged 10000x $g_n$ for 90 min at 4˚C. The 500 $\mu L$ of plasma filtrate was mixed with NMR buffer up to 600 uL.

**NMR spectroscopy acquisition and processing.** The experiments are conducted using 200 μL cell and 550 μL media samples in 103.5 mm x 3 mm and 103.5 mm x 5 mm NMR tubes (Bruker). All the data collection and processing were performed using Topspin 3.6 software (Bruker Analytik, Rheinstetten, Germany). One-dimensional [1]H NMR spectra are acquired on a Bruker Avance II 600 MHz spectrometer using Prodigy BBO cryoprobe at 298 K using the noesygppr1d pulse sequence (Ref1). For a representative sample, two dimensional data [1]H-[1]H total correlation spectroscopy (TOCSY, mlevphpr.2) and 2D [1]H-[13]C heteronuclear single quantum coherence (HSQC, hsqcedetgpsisp2.2) were collected for metabolites assignment.

**Metabolites assignments and quantification.** Metabolites found in cell extract are assigned based on 1D [1]H and 2D NMR experiments. Peaks are assigned by comparing the chemical shifts and spin-spin couplings with reference spectra found in databases, such as the Human Metabolome Database (HMDB) (Ref2), and Chenomx[®] NMR Suite profiling software (Chenomx Inc. version 8.1). The concentrations of the metabolites are calculated using Chenomx software based on the internal standard, TMSP.

## Results

### Chronic Her2 inhibition leads to Her3-dependent reactivation of mitogenic signaling

To study the kinetics of proteostatic perturbations during the cellular adaptation to chronic Her2 inhibition, we followed a frequently employed *in vitro* approach to the modeling of acquired resistance by prolonged incubation of Her2-amplified SKBR3 breast cancer cells in

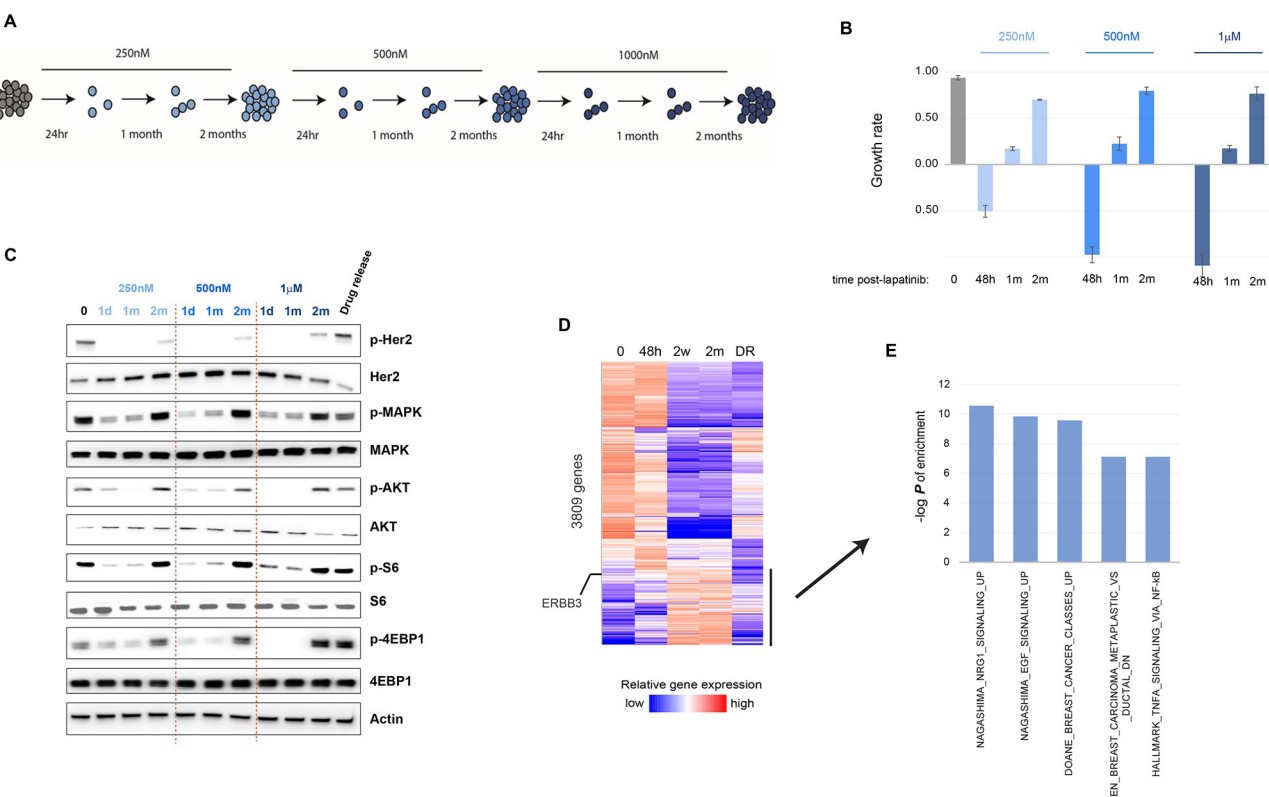

**Fig 1. Developing a model of acquired resistance to anti-Her2 therapy *in vitro*.** A) SKBR3 breast cancer cells were cultured in lapatinib till they gained resistance and started active growth under drug (typically ∼ 2 months), when the lapatinib concentration was increased as shown. B) Growth rate (difference in relative viability between days 1 and 3 of seeding divided by 2, see Methods) at the various stages of chronic lapatinib treatment. C) Western blot of the indicated phospho- and total proteins for key mitogenic signaling pathways downstream of Her2 at the indicated time points after the start of lapatinib treatment. The "Drug release" time point indicates 48 hours after the removal of the drug from the media after the 2 months in lapatinib. D) Whole-transcriptome analyses of cells at the indicated stages of 1μM lapatinib treatment. Coloring reflects z-score of normalization across the conditions for each gene. The highlighted portion of the heatmap shows genes that are selectively upregulated at the resistance (2W: 2 weeks) and relapse (2M: 2 months) phases. E) The signatures that were most enriched for the highlighted genes in (D) based on GSEA analysis. Statistics: In (B), error bars show standard deviation of 3 replicates, and is representative of multiple (>2) independent experiments. Densitometric quantitation of the immunoblotting data are provided in S1 Table.

increasing doses of lapatinib (Fig 1A). After an initial period of drug-induced cytotoxicity, cells reach stasis (resistance phase) after 2 weeks of drug exposure, and start active growth in drug after about 2 months (relapse phase), at which point we repeated the cycle with a higher drug dose, until cells were actively growing in 1μM lapatinib (Fig 1A and 1B). As expected, resistance and relapse on lapatinib were associated with the activation of the Akt, MAP and mTOR kinases during chronic lapatinib treatment (Fig 1C).

RNAseq-based transcriptomic profiling of cells at each stage of tumor cell growth under 1μM lapatinib, followed by gene set enrichment analysis of corresponding signatures, revealed that the resistance and relapse phases display a signature consistent with the activation of neuregulin (NRG1) and EGF signaling (Fig 1D and 1E). Neuregulin is a ligand for Her3 receptor tyrosine kinase, which dimerizes with Her2 to promote mitogenic signaling, and whose overexpression mediates acquired resistance to Her2 inhibition [17–20]. Accordingly, total and phosphorylated forms of Her3 levels were dramatically induced shortly after lapatinib treatment (Fig 2A). Consistent with its central role in the acquired resistance to Her2 inhibition, the activation of the downstream pathways, and hence survival under lapatinib, were strongly dependent on Her3 (Fig 2B–2D). Accordingly, while the knock-down of Her3 prevented, its

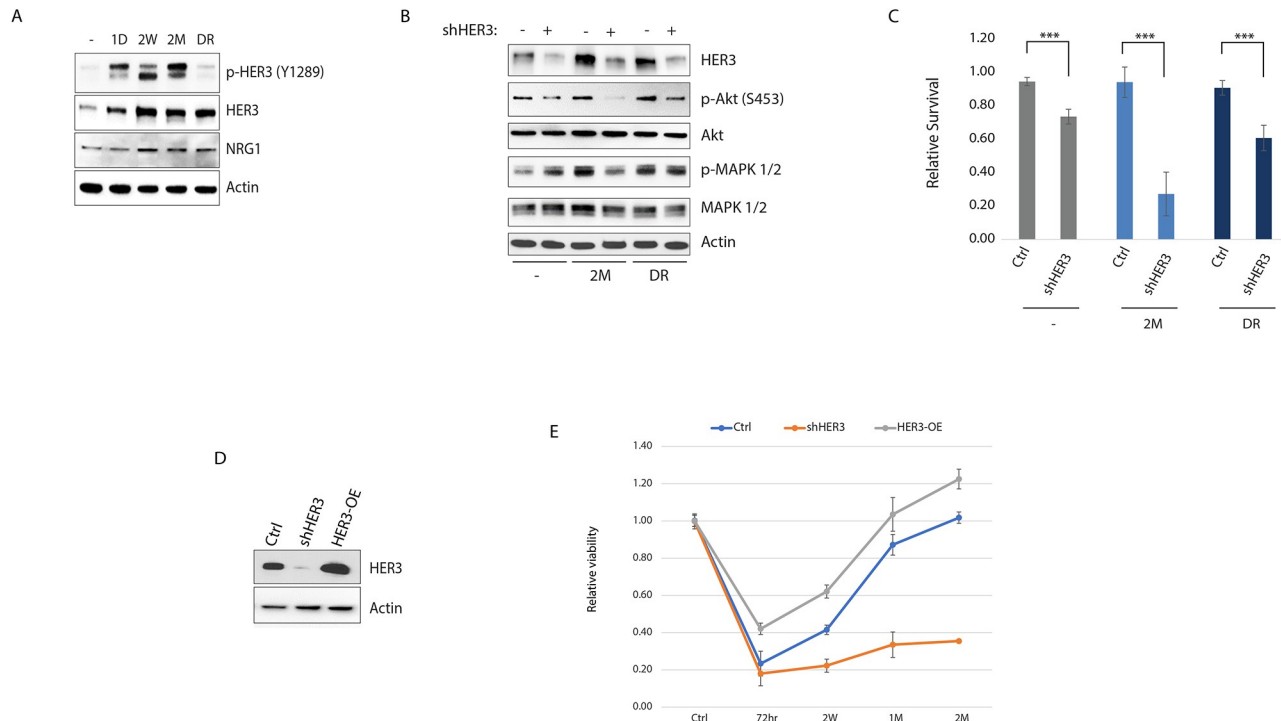

**Fig 2. Her3 overexpression confers tumor cell growth under lapatinib.** A) Total and phospho-Her3 levels at the indicated stages of lapatinib ($1\mu M$) treatment. B) Western blot of the indicated signaling proteins with and without shRNA-mediated knock-down of Her3 at the indicated stages of lapatinib treatment. C) Relative viability (compared to control) under the same conditions. D) Her3 was stably silenced or overexpressed in SKBR3 cells, and E) the relative cell growth (relative to control) was calculated after $1\mu M$ lapatinib treatment for the indicated time period. Statistics: error bars show standard deviations of 6 (C) or 3 (E) replicate experiments. ***: $P < 0.01$ with student's t-test. Densitometric quantitation of the immunoblotting data are provided in S1 Table.

overexpression facilitated, the acquired resistance to lapatinib (Fig 2E), confirming the previously established role of Her3 as a substitute oncogenic kinase for Her2. Interestingly, this oncogene switch was highly dynamic, as release of cells from lapatinib at the relapse phase (drug release phase: 48 hours in drug-free media) restored total and phosphorylated Her2 levels and reversed the upregulation of phospho-, but not total, levels of Her3 in a few days, and reversed most of the gene expression changes associated with chronic lapatinib treatment, with little effect on the downstream signaling pathways (Figs 1C and 2A–2C).

## Chronic Her2 inhibition leads to proteostatic perturbation and ER stress

We previously showed that the survival of Her2+ breast cancer cells is critically dependent on balancing the cell's protein folding capacity with its protein synthesis load [12]. To study how cellular protein homeostasis is modulated during chronic Her2 inhibition, we measured the protein synthesis rates during each phase of drug inhibition. Not surprisingly, protein synthesis rates closely followed cell growth kinetics, with a dramatic acute inhibition of protein synthesis in the initial phase, followed by stasis in the resistance, and complete recovery in the relapse phase (Fig 3A and 3B). The recovery of protein synthesis during the relapse phase was dependent on Her3, but this dependence on Her3 diminished shortly after release from lapatinib (Fig 3C). As we showed previously, Her2 inhibition in the acute phase resulted in ER stress characterized by phosphorylation of PERK, and of its target eIF2-α on Ser51, a hallmark of ER stress-induced inhibition of protein synthesis (Fig 3D). Interestingly, activation of Her3 and of downstream signaling pathways in the resistance phase did not alleviate, but further

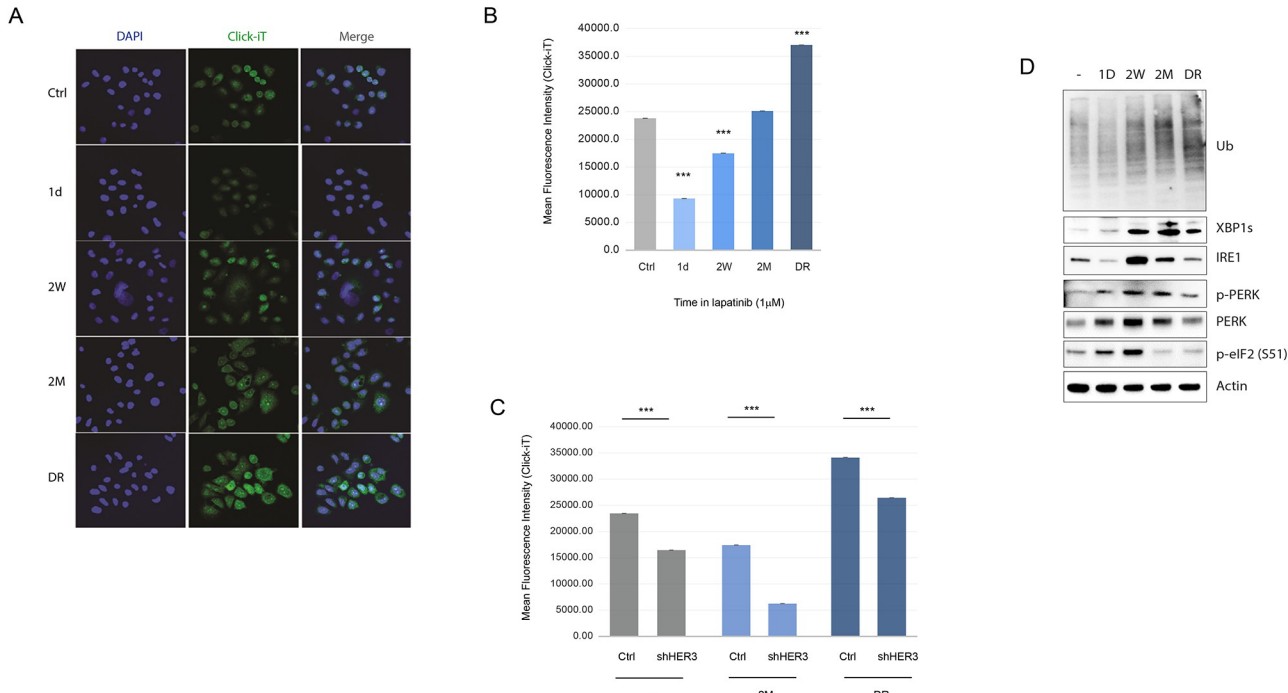

**Fig 3. Protein homeostasis dynamics during the course of chronic lapatinib treatment.** A) Immunofluorescence images of at the indicated stages of 1$\mu$M lapatinib treatment stained for newly synthesized proteins within a 30 min window using the Click-iT Protein Synthesis assay. B) Flow cytometry-based measurement of protein synthesis using the same conditions as in A using the Click-iT assay. Data is presented as mean fluorescence intensity (MFI). C) Protein synthesis rate measurement as in (B) in the indicated conditions with and without Her3 silencing. D) Western blots of the indicated ER stress response markers in the indicated conditions. Statistics: error bars show standard deviations of 2 (B-C) replicates. The data are representative of at least 2 independent experiments. ***: $P < 0.01$ with student's t-test. Densitometric quantitation of the immunoblotting data are provided in S1 Table.

exacerbate, the ER stress phenotype (Fig 3D), suggesting that Her3 activation is unable to suppress the ER stress phenotype induced by Her2 inhibition. However intriguingly, the relapse phase and the recovery of Her3-mediated protein synthesis and growth was characterized by the alleviation of eIF2 inhibitory phosphorylation on Ser51, despite the strong persistence of high levels of phosphorylated PERK (Fig 3D), indicating an uncoupling of PERK from eIF2 phosphorylation. Interestingly, the drug release phase was characterized by an even higher protein synthesis load (Fig 3A), indicating a surge in the protein synthesis rates caused by Her2 de-inhibition. Moreover, despite Her2 re-activation, the increased XBP1s and reduced p-eIF2 levels persisted at the drug release stage (Fig 3D).

## Chronic Her2 inhibition blocks glycolysis and forces shift to oxidative metabolism

Next, we sought to gain insight into the mechanisms of ER stress during the different phases of Her2 inhibition. We and others have reported that acute Her2 inhibition impairs glucose uptake, which contributes to ER stress [21]. To analyze the dynamics of metabolic pathway activity changes during the acute and chronic drug response in Her2+ cells, we performed NMR-based intracellular (Fig 4A) and extracellular (S3A Fig) metabolomics profiling of cells at each phase of the drug response cycle. As expected, the remission phase was characterized by the increases in glucose and amino acid levels intracellularly (Fig 4A), consistent with their reduced breakdown, as well as extracellularly (S3A Fig), consistent with the drop in their

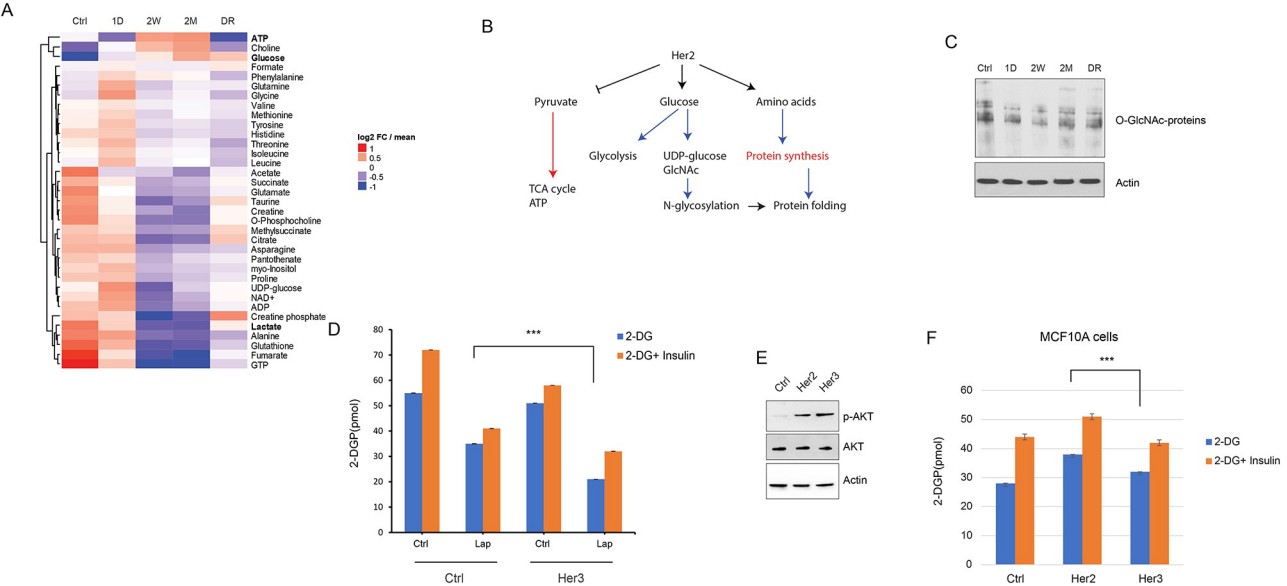

**Fig 4. Metabolomic reprogramming during chronic lapatinib treatment.** A) Heatmap of the most notable altered intracellular metabolites between the different stages of 1$\mu$M lapatinib treatment, measured by NMR mass spectrometry. B) A model summarizing the effect of chronic lapatinib treatment on the glucose, energy and amino acid metabolism. Blue lines: reduced flux, red lines: increased flux at the relapse stage (2M). C) Western blot of O-GlcNAc-conjugated protein levels in the indicated conditions. O-GlcNAc levels may serve as a readout of the N-linked hexosamine (GlcNAc) levels. D) Intracellular glucose uptake assay measuring 2-deoxyglucose uptake with and without insulin after lapatinib treatment in SKBR3 cells. Her3 overexpression is unable to rescue the glucose uptake inhibition of lapatinib treatment. E) Overexpression of Her2 or Her3 in the non-transformed MCF10A cells induces similar activation of downstream pathways (p-Akt). F) Glucose uptake rates with control, Her2 or Her3 overexpression in MCF10A cells. Statistics: error bars show standard deviations from 2 replicate samples. Data are representative of 2 independent experiments. ***: $P < 0.01$ with student's t-test. Densitometric quantitation of the immunoblotting data are provided in S1 Table.

cellular uptake. In addition, there was a significant reduction in the intracellular (Fig 4A) and extracellular lactate levels (S3 Fig), along with a drop in ATP and an increase in ADP levels, consistent with a global reduction in glycolysis and energy metabolism. Significant increases in the intracellular and extracellular amino acid pools, in turn, is consistent with the global reduction in the protein synthesis (see Fig 3), leading to reduced uptake and utilization of amino acids in protein synthesis.

Interestingly, the resistance and the relapse phases had highly similar profiles to each other, with significant increase in ATP generation and intracellular glucose, and a decrease in the lactate and amino acid levels, as well as in the citric acid (TCA) cycle intermediates such as citrate, succinate and fumarate. These observations suggest that, despite resumption of ATP generation at an even higher rate compared to the basal state of these cells (see Fig 4A), and resumption of protein synthesis (i.e. consumption of intracellular amino acid pools), flux through glycolysis remains low. However, there may be a switch towards the increased utilization of TCA cycle and mitochondrial oxidative phosphorylation for ATP generation as the cells reprogram their metabolism during adaptation to chronic Her2 inhibition (Fig 4B), similar to what has been reported previously in other contexts of acquired drug resistance [22–26].

The inhibition of glucose uptake during acute and chronic Her2 inhibition was accompanied by reduced flux through the nucleotide and amino-sugar pathway, as evidenced by reduced levels of UDP-glucose (see Fig 4A) and O-linked acetyl-glucosamine, despite their slight restoration at the relapse stage (Fig 4C). These observations suggest that the compensatory activation of Her3, while sufficient to reactivate the downstream signaling pathways and protein synthesis, is not sufficient to rescue the metabolic defects of Her2 inhibition, which probably culminates in ER stress. Accordingly, the overexpression of Her3 was not

able to restore glucose uptake defects in lapatinib-treated SKBR3 cells (Fig 4D), and was less efficient in triggering glucose uptake in non-transformed mammary epithelial cells (MCF10A) compared to Her2, despite being equally capable of activating the downstream Akt phosphorylation (Fig 4E and 4F). Interestingly, release of cells from lapatinib at the relapse stage leads to the partial resumption of glycolysis (evidenced by lactate production) and a partial shut-down of TCA (evidenced by increased citrate and reduced ATP). The levels of creatine phosphate, a product of the shuttle between the mitochondria and the cytosol to buffer high-energy phosphate in the cell, also strongly follows a similar pattern, perhaps reflecting the switch in mitochondrial metabolism, or even direct Her2 activity as the creatine phosphate pathway has been recently shown to be directly regulated oncogenic Her2 [27]. The levels of free amino acids are even further reduced, consistent with a surge in the protein synthesis rates upon de-inhibition of Her2 (see Fig 4A). These observations suggest that while Her3 activation during chronic Her2 inhibition is able to restore mitogenic signaling, it is not able to sustain a glycolytic phenotype in the absence of Her2, forcing cells to utilize mitochondrial OXPHOS.

## Overexpression of GADD34 allows protein synthesis and cell growth under ER stress

To identify the factors that mediate tumor cell growth during glucose starvation and subsequent ER stress under chronic lapatinib treatment, we analyzed the genes that were specifically overexpressed (z-score of $> 1$) in the relapse, but not any other, condition. Interestingly, applying this filter only revealed several genes, one of which was *PPP1R15A*, a gene that encodes the GADD34 subunit of the protein phosphatase 1 (PP1) (S1 Fig). Importantly, one of the best-characterized functions of GADD34 is to mediate the dephosphorylation of Ser51 on eIF2α by PP1, reversing PERK-mediated phosphorylation and inhibition of eIF2α activity [27].

Phosphorylation of eIF2α on Ser51 by PERK is an essential step to inhibit protein synthesis during ER stress to prevent the accumulation of misfolded proteins [28, 29]. Recovery of protein synthesis and cell growth during the resistance phase of chronic Her2 inhibition is accompanied by the loss of Ser51 phosphorylated eIF2α despite the persistence of active ER stress response and active PERK (see Fig 3D). GADD34 is one of the PP1 subunits that mediate eIF2α dephosphorylation, and is known to be responsive to ER stress [27–30]. Indeed, GADD34 expression is dramatically induced at both mRNA and protein levels starting in the resistance phase and reaching peak levels during the relapse and drug release phases, where eIF2α dephosphorylation takes place (Fig 5A). Importantly, inhibiting GADD34 by treatment with guanabenz, a specific inhibitor of GADD34 [31], or by GADD34-targeting shRNA restored eIF2α phosphorylation, protein synthesis block and growth inhibition at the relapse-stage cells (Fig 5B–5G, S3B Fig), suggesting that GADD34 overexpression during the resistance and relapse phases allows cells to uncouple the metabolic ER stress response from protein synthesis to permit active growth (Fig 5H).

To test if *PPP1R15A*/GADD34 expression correlates with anti-Her2 therapy response in human patients, we analyzed the transcriptomic data from the Long-HER study, which obtained whole-genome gene expression measurements in advanced Her2+ breast cancer patients who had long ($>3$ years) durable response to first-line trastuzumab therapy, relative to the group who had a poor response ($<1$ year response) [32]. Importantly, in this cohort, high expression of PPP1R15A significantly correlated with the shorter duration of response (Fig 5I), strongly suggesting a role for the GADD34—eIF2 axis in the resistance to anti-Her2 therapy in the clinic.

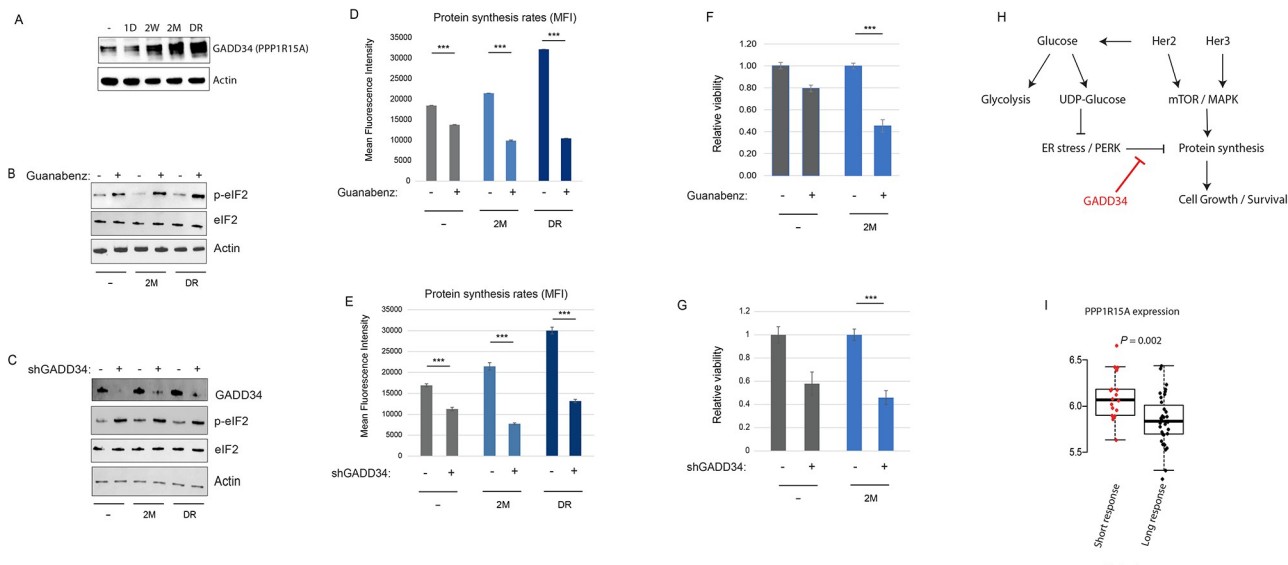

**Fig 5. GADD34 overexpression during prolonged lapatinib treatment allows to overcome ER stress response-mediated inhibition of protein synthesis.** A) Protein levels of GADD34 (PPP1R15A) in the indicated conditions. B-C) Western blot of p-eIF2 (S51) under indicated conditions with and without treatment with guanabenz (GADD34 inhibitor) (B) or GADD34-targeting shRNA (C). D-E) Relative viability of cells at the control (parental) and relapse (2M) stages in response to 72hr treatment with guanabenz (D) or shRNA against GADD34 (E). F-G) Protein synthesis rate measurement with Click-iT Protein synthesis kit in the indicated stages of lapatinib treatment with and without guanabenz (F) or shRNA targeting GADD34 (G). Data is presented as mean fluorescence intensity (MFI). H) A model summarizing the signaling, metabolic and proteostatic changes during the acquisition of resistance to lapatinib in SKBR3 cells. I) mRNA levels of PPP1R15A in the cohort of advanced Her2+ breast cancer patients (n = 53 samples) with short (<1 year) or long (>3 years) duration of response to first-line trastuzumab therapy, from the Long-HER study (33). Statistics: Error bars show standard deviations of 2 (D-E) or 3 (F-G) replicate samples. Data are representative of at least 2 independent experiments (D, F). ***: $P < 0.01$ with student's t-test. Densitometric quantitation of the immunoblotting data are provided in S1 Table.

## Cells at the relapse and drug release stages are vulnerable to ERAD inhibition

We have shown that increased proteotoxic load in Her2+ breast cancer cells creates a dependency on the ER-associated degradation (ERAD) pathway to prevent cytotoxic ER stress [12]. We asked if the proteotoxic state induced by the oncogenic switch from Her2 to Her3 during chronic Her2 inhibition and later drug release creates a similar, or more pronounced, dependence on ERAD for survival. Intriguingly, ablation of expression of Valosin-containing Protein (VCP), the core ATPase of the VCP/UFD1L/NPL complex responsible for the extraction and the delivery of the misfolded proteins to the proteasome [33], has significantly higher toxicity to cells in the relapse phase, and even higher toxicity at the drug release phase (Fig 6A–6C). This was accompanied by significant accumulation of poly-ubiquitinated proteins in these phases of drug response, indicating a heightened proteotoxic state. The use of a recently developed specific VCP inhibitor [34] resulted in a similar phenotype (Fig 6D and 6E), especially more evident at a higher dose, suggesting that the vulnerability of the relapse and drug release phases of chronic Her2 inhibition is pharmacologically exploitable.

In order to test if the mechanisms of resistance to Her2-targeted therapy and the molecular vulnerabilities thereof presented above are applicable to other models, we carried out a similar approach using another Her2+ breast cancer cell line model, BT474. Interestingly, continuous incubation of BT474 cells in increasing doses of lapatinib allowed them to actively grow in 400nM of the drug (S2A Fig), associated with concomitant activation of Her3 and the downstream pathways (S2B Fig). In addition, similar to the SKBR3 model, active growth in lapatinib in BT474 cells was associated with the ER stress, a dramatic increase in GADD34 levels, and

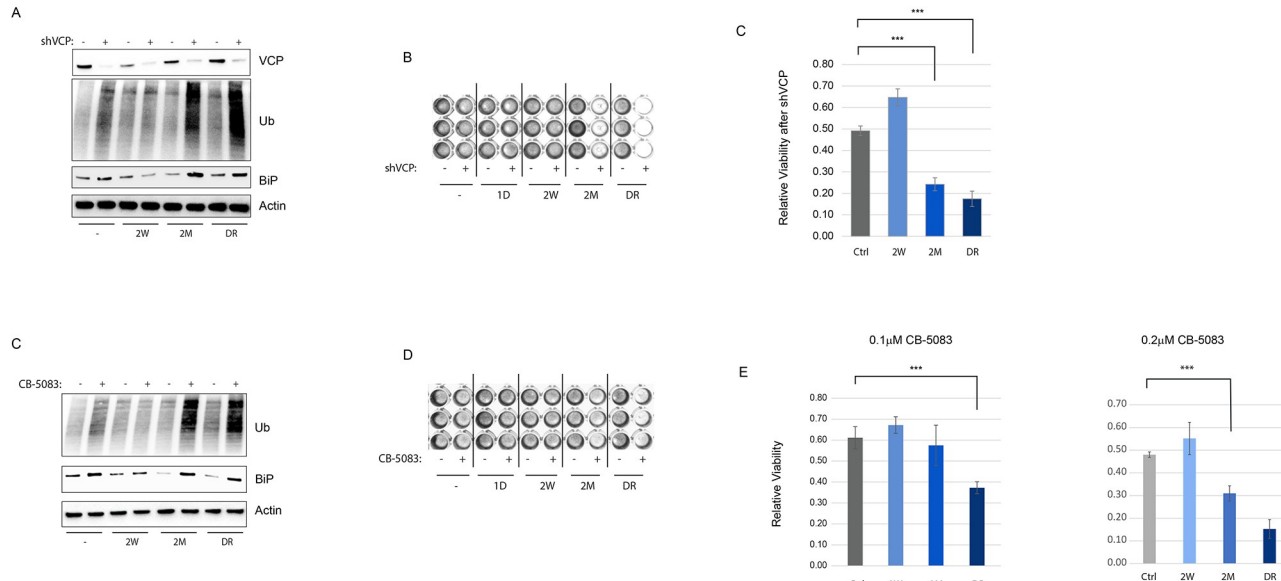

**Fig 6. Acquired resistance to lapatinib creates hyper-dependence on ERAD for survival.** A) Western of Ub (for poly-ubiquitinated proteins) and BiP (ER stress response marker) after silencing of p97 VCP in the indicated stages of lapatinib treatment. B) Images of Crystal Violet stained wells of 96-well plate after 72hrs of silencing of VCP in cells of the indicated stages of lapatinib exposure. C) Quantitation of the relative intensities of readings in (B). D-E) Similar to A,C but using 0.1 and 0.2$\mu$M treatment with CB-5083, a highly selective clinical grade inhibitor of VCP. Statistics: error bars show standard deviations from 6 replicate samples. Data are representative of at least 2 independent experiments. ***: $P < 0.01$ with student's t-test.

the corresponding decrease in eIF2 phosphorylation (S2C Fig). While acute lapatinib treatment was associated with decreased protein synthesis, the relapse phase was associated with a significantly increased protein synthesis, which was reversible with guanabenz treatment (S2D Fig). Finally, cells at the relapse phase were more sensitive to the VCP inhibitor CB-5083 (S2E Fig). The results from our orthogonal BT474 model, therefore, support the notion that the GADD34-eIF2 axis plays an important role in Her3-mediated growth during chronic Her2 inhibition, and that the resulting imbalance in the ER proteotoxic load sensitizes cells to the inhibitors of the ER protein clearance pathways.

## Discussion

Targeted inhibition of oncogenic kinases is a promising therapeutic option in several molecularly defined contexts, and mAb-based targeting of the Her2 oncogene in breast cancers has dramatically changed the outcome of this subclass of the disease. However, *de novo* and acquired resistance to kinase inhibitors is a major barrier to therapeutic success, especially in the advanced setting [5]. Transcriptional and post-transcriptional overexpression of the pseudo-kinase Her3 and subsequent hetero-dimerization with Her2 has been consistently shown as a major mediator of acquired resistance to Her2 inhibition, driving Her2-independent activation of the downstream mitogenic pathways [17–20]. Several other mechanisms, most involving alternative receptor tyrosine kinases, have been proposed as mechanisms of acquired resistance to Her2 inhibition in different cell models [35], underscoring the multitude of "bypass" options [36] that are available for the tumor cell to evade therapy. Indeed, widespread "adaptive kinome reprogramming" in response to chronic Her2 inhibition has been reported to upregulate a variety of compensatory receptor tyrosine kinases, each with the demonstrated ability to independently activate the downstream oncogenic pathways (mainly PI3K/mTOR) and tumor growth [37]. Unfortunately, clinical trials for combination

treatments with Her2 and mTOR pathways in advanced Her2+ breast cancers, despite evidence for prolonged progression-free survival [38, 39], have not yielded changes to the standard treatment due to significant toxicity associated with the combination [5], underscoring the need for alternative targeting strategies.

A relatively understudied concept in the field is exploiting the therapeutically targetable vulnerabilities imposed on the cell by oncogenic activation, or oncogenic stress. The central theme in this concept is that the functional state of the tumor cell gained through evolutionary adaptation to the oncogenic burden or chronic drug treatment is highly fragile and can be targeted by identifying these vulnerability points. These so-called non-oncogene addictions [6] or collateral sensitivities [40] in cancer cells have been shown to be viable therapeutic strategies for the killing of tumor cells with non-targetable oncogenes (e.g. *MYC*) [10] or those that have gained resistance to targeted therapy [13, 40]. For example, MYC- and KRAS-driven cancers have been reported to be particularly sensitive to the inhibitors of the key players of lipid and glucose metabolism due to the oncogenic reprogramming of the cellular metabolic pathways [41–45]. Similarly, the proteostatic imbalance created by oncogenic transformation has also been shown to create vulnerabilities within the adaptive proteotoxic stress response pathways, presenting further opportunities for therapeutic exploitation [11, 46].

Along this line, we previously reported that Her2+ breast cancer cells have an acute dependence on the ER-associated degradation pathway for survival, due to the severe proteotoxic stress imposed by the genomic amplification and hyper-active signaling by the Her2 oncogene [12]. In the present study, we asked if a similar dependency exists in Her2+ breast cancer cells within the context of acquired resistance to Her2 inhibition, as this is the most likely clinical context where ERAD-targeted therapy might be considered in this group of patients. Moreover, it is not clear how the signaling and protein homeostasis dynamics change during the course of chronic anti-Her2 therapy, and what alternative vulnerabilities might be imposed by the very mechanisms that confer resistance to Her2 inhibition.

Our study using the *in vitro* drug dosing approach of acquired resistance to Her2 inhibition reproduces the central role of Her3 in taking over the mitogenic signaling in Her2 + breast cancer cells (Figs 1 and 2). However, intriguingly, the overexpression of Her3 is unable to compensate for the glucose uptake and ER proteostasis defects of chronic Her2 inhibition (Figs 3 and 4), thus resulting in major metabolic remodeling in the resistant cells (Fig 4). The uncoupling of the ER stress response from the regulation of protein synthesis is accomplished by the dramatic overexpression of GADD34, a subunit of PP1 complex, which mediates the dephosphorylation of the Ser51 on eIF2α, allowing cells to resume protein synthesis and growth despite the active ER stress response (Fig 5). Intriguingly, active protein synthesis and cell growth under such unresolved ER stress further strains the protein homeostasis machinery, creating a hyper-dependence on the ER quality control system, which includes ERAD, to mitigate the proteotoxic imbalance and sustained viability (Fig 6). It is important to note that the high expression of GADD34 correlates with a shorter duration of response to anti-Her2 therapy in breast cancer patients in the clinic (Fig 5H), suggesting that our findings from our *in vitro* model reveal clinically relevant mechanisms, and that targeting of the ER quality control system might be a viable therapeutic option post-progression on Her2 inhibitors.

## Supporting information

**S1 Table. List of antibodies.**
(XLSX)

**S2 Table. RNAseq data.**
(XLSX)

**S1 Fig. Expression profiles of genes with the most overexpression in the relapsed samples.**
A) heatmap of top genes whose expression was increased in the relapse (2M) samples (from
RNAseq). B) Bar plot of expression values for PPP1R15A (from RNAseq) in the indicated sam-
ples.
(PDF)

**S2 Fig. The GADD34-eIF2 axis plays an important role in Her3-mediated growth during
chronic Her2 inhibition in BT474 cells.**
(PDF)

**S3 Fig. Extracellular metabolomics profiling of SKBr3 cells at different phases of the drug
response cycle.**
(PDF)

**S1 Raw images. Raw blot images.**
(PDF)

## Acknowledgments

We would like to acknowledge the assistance of the Research Flow Cytometry Core in the
Division of Rheumatology and the NMR-based Metabolomics Core Facility at Cincinnati Chil-
dren's Hospital Medical Center.

## Author Contributions

**Conceptualization:** Lisa M. Privette Vinnedge, Kakajan Komurov.

**Data curation:** Lindsey Romick-Rosendale.

**Formal analysis:** Navneet Singh.

**Funding acquisition:** Kakajan Komurov.

**Investigation:** Navneet Singh, Lindsey Romick-Rosendale, Miki Watanabe-Chailland, Lisa M.
Privette Vinnedge.

**Project administration:** Kakajan Komurov.

**Supervision:** Lisa M. Privette Vinnedge, Kakajan Komurov.

**Validation:** Navneet Singh.

**Writing – original draft:** Lisa M. Privette Vinnedge, Kakajan Komurov.

**Writing – review & editing:** Kakajan Komurov.

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
