## [Decision Letter · Decision Letter 0]

8 Sep 2021

PONE-D-21-25753Drug resistance mechanisms create targetable proteostatic vulnerabilities in Her2+ breast cancersPLOS ONE

Dear Dr. Komurov,

Thank you for submitting your manuscript to PLOS ONE. After careful consideration, we feel that it has merit but does not fully meet PLOS ONE’s publication criteria as it currently stands. Therefore, we invite you to submit a revised version of the manuscript that addresses the points raised during the review process. Please respond to the comments item-by-item to satisfy the reviewers' concerns. Please carefully edit the MS for English corrections and typos. Please submit your revised manuscript by Oct 23 2021 11:59PM. If you will need more time than this to complete your revisions, please reply to this message or contact the journal office at plosone@plos.org. Please include the following items when submitting your revised manuscript:A rebuttal letter that responds to each point raised by the academic editor and reviewer(s). You should upload this letter as a separate file labeled 'Response to Reviewers'.A marked-up copy of your manuscript that highlights changes made to the original version. You should upload this as a separate file labeled 'Revised Manuscript with Track Changes'.An unmarked version of your revised paper without tracked changes. You should upload this as a separate file labeled 'Manuscript'.

We look forward to receiving your revised manuscript.

Kind regards,

Nandini Dey, MS., Ph.D

Academic Editor

PLOS ONE

Journal Requirements:

2. In your Methods section, please provide additional details regarding the cell lines used in your study and ensure you have described the source. For more information regarding PLOS' policy on materials sharing and reporting, see https://journals.plos.org/plosone/s/materials-and-software-sharing#loc-sharing-materials, and for more information on PLOS ONE's guidelines for research using cell lines, see https://journals.plos.org/plosone/s/submission-guidelines#loc-cell-lines.

"This work was supported by NIH awards R01CA193549 (KK and LMPV), R37CA218072 (LMPV), and a Department of Defense Breast Cancer Research Program level I award W81XWH-16-1-0028 (NS). We would like to acknowledge the assistance of the Research Flow Cytometry Core in the Division of Rheumatology and the NMR-based Metabolomics Core Facility at Cincinnati Children’s Hospital Medical Center."

"This work was supported by NIH awards R01CA193549 (KK and LMPV), R37CA218072 (LMPV), and a Department of Defense Breast Cancer Research Program level I award W81XWH-16-1-0028 (NS). We would like to acknowledge the assistance of the Research Flow Cytometry Core in the Division of Rheumatology and the NMR-based Metabolomics Core Facility at Cincinnati Children’s Hospital Medical Center. The funders had no role in study design, data collection and analysis, decision to publish, or preparation of the manuscript."

6. Thank you for stating the following in the Competing Interests section: 

"KK is an employee of Champions Oncology Inc, and holds stocks there and at Pfizer Inc."

We note that you received funding from a commercial source: Champions Oncology Inc, and Pfizer Inc.

7. In your Data Availability statement, you have not specified where the minimal data set underlying the results described in your manuscript can be found. PLOS defines a study's minimal data set as the underlying data used to reach the conclusions drawn in the manuscript and any additional data required to replicate the reported study findings in their entirety. All PLOS journals require that the minimal data set be made fully available. For more information about our data policy, please see http://journals.plos.org/plosone/s/data-availability.

8. PLOS ONE now requires that authors provide the original uncropped and unadjusted images underlying all blot or gel results reported in a submission’s figures or Supporting Information files. This policy and the journal’s other requirements for blot/gel reporting and figure preparation are described in detail at https://journals.plos.org/plosone/s/figures#loc-blot-and-gel-reporting-requirements and https://journals.plos.org/plosone/s/figures#loc-preparing-figures-from-image-files. When you submit your revised manuscript, please ensure that your figures adhere fully to these guidelines and provide the original underlying images for all blot or gel data reported in your submission. See the following link for instructions on providing the original image data: https://journals.plos.org/plosone/s/figures#loc-original-images-for-blots-and-gels. 

9. Please amend the manuscript submission data (via Edit Submission) to include author  Lindsey Romick-Rosendale and Miki Watanabe-Chailland.

10. Please include captions for your Supporting Information files at the end of your manuscript, and update any in-text citations to match accordingly. Please see our Supporting Information guidelines for more information: http://journals.plos.org/plosone/s/supporting-information. 

Reviewers' comments:

Reviewer's Responses to Questions

**Comments to the Author**

1. Is the manuscript technically sound, and do the data support the conclusions?

Reviewer #1: Yes

Reviewer #2: Partly

2. Has the statistical analysis been performed appropriately and rigorously? 

Reviewer #1: N/A

Reviewer #2: Yes

3. Have the authors made all data underlying the findings in their manuscript fully available?

Reviewer #1: Yes

Reviewer #2: No

4. Is the manuscript presented in an intelligible fashion and written in standard English?

Reviewer #1: Yes

Reviewer #2: Yes

5. Review Comments to the Author

Reviewer #1: In the era/context of precision medicine, oncogenic kinase inhibitor is the key for drug matching with specific oncogenic mutations but their response often short lived due to development of clinical (ad inito or acquired ) resistance. In this aspect this is an interesting/important article.

Critical comments:

1. This is a question to authors: Can this proteostatic vulnerabilities correlate well with immune vulnerabilities following the treatment of ant-HER2 therapies? Since authors demonstrated some correlation between proteostatic and genomic vulnerabilities.

2. Authors performed some experiments with BT474 cells (along with SRBR3 cells), both cells are HER2 amplified, but SKBR3 is ER negative and BT474 is ER+ cells. In their hands both behave almost similarly following the development of lapatinib resistance. Any comments from authors side about the role of ER in the development of lapatinib-induced metabolic resistance.

3. Authors may provide NRG1 and EGF Western blot expression data.

4. In Figure 4A, please make it bold for UDP-glucose (in the figure).

5 In line 290, where authors mentioned increased citrate expression, here authors failed to mention about the high expression of Creatin Phosphate.

6. Authors may provide qRT-PCR data of GADD34 mRNA expression.

7. Please mentioned number of patients sample for Figure 5H. These patients were exposed with trastuzumab but not lapatinib. Any thoughts from authors side?

Minor comments:

1. Authors failed to mentioned about the phosphorylation sites of AKT following lapatinib treatment. Better to provide both Ser473 and Thr308 phosphorylation expression.

Reviewer #2: The Authors, in their work, using transcriptomic, metabolomic and proteomic analysis, characterized the mechanism of acquisition of drug resistance in Her2+ breast cells. The topic of the manuscript is attractive and very important; however, some points need clarification:

• Transcriptomics and qPCR methods are not described in the material and methods section.

• The raw data of transcriptomic and metabolomic analysis should be registered in the public repository, and the accession ID should be displayed in the main text. Moreover, it would be helpful to include the results of these analyses (lists of genes and metabolites with most altered expression) as tables in supplementary materials.

• The results chapter should be divided into subsections that would make this chapter clearer.

• There are no units in figures 1B, 2C, 2E, 3B, 3C, 4C, 4E, and supplementary results.

• The Authors should explain how the relative viability and growth rate were counted (presented in figures).

• What was the reference gene used to normalize the qPCR results?

• The discussion in this publication is too general. The Authors should discuss their results in the context of other published work instead of just repeating the description of their results.

• There are some grammar and spelling mistakes.

6. PLOS authors have the option to publish the peer review history of their article (what does this mean?). If published, this will include your full peer review and any attached files.

Reviewer #1: No

Reviewer #2: No

---

## [Author Response · Author response to Decision Letter 0]

10 Feb 2022

The point-by-point rebuttal letter has been attached.

---

## [Editor Report · Decision Letter 1]

23 Feb 2022

Drug resistance mechanisms create targetable proteostatic vulnerabilities in Her2+ breast cancers

PONE-D-21-25753R1

Dear Dr. Komurov,

We’re pleased to inform you that your manuscript has been judged scientifically suitable for publication and will be formally accepted for publication once it meets all outstanding technical requirements.

Kind regards,

Nandini Dey, MS., Ph.D

Academic Editor

PLOS ONE
---

## [Editor Report · Acceptance letter]

14 Mar 2022

PONE-D-21-25753R1 

Drug resistance mechanisms create targetable proteostatic vulnerabilities in Her2+ breast cancers 

Dear Dr. Komurov:

I'm pleased to inform you that your manuscript has been deemed suitable for publication in PLOS ONE. Congratulations! Your manuscript is now with our production department. 

Kind regards, 

on behalf of

Dr. Nandini Dey 

Academic Editor

PLOS ONE